# Searching Anti-Zika Virus Activity in 1*H*-1,2,3-Triazole Based Compounds

**DOI:** 10.3390/molecules26195869

**Published:** 2021-09-28

**Authors:** Willyenne M. Dantas, Valentina N. M. de Oliveira, Diogo A. L. Santos, Gustavo Seabra, Prem P. Sharma, Brijesh Rathi, Lindomar J. Pena, Ronaldo N. de Oliveira

**Affiliations:** 1Department of Chemistry, Federal Rural University of Pernambuco, Recife 52171-900, Brazil; willyenne.dantas@ufrpe.br; 2Department of Virology, Aggeu Magalhães Institute (IAM), Oswaldo Cruz Foundation (Fiocruz), Recife 50670-420, Brazil; 3Instituto Federal de Educação Ciência e Tecnologia de Pernambuco, Campus Ipojuca, Ipojuca 55590-000, Brazil; valentinamelo@ipojuca.ifpe.edu.br; 4Department of Fundamental Chemistry, Federal University of Pernambuco, Recife 50740-540, Brazil; diogoalveslopessantos@gmail.com (D.A.L.S.); gustavo.seabra@gmail.com (G.S.); 5Laboratory for Translational Chemistry and Drug Discovery, Department of Chemistry, Hansraj College, University of Delhi, Delhi 110007, India; premprakash.cr@gmail.com (P.P.S.); brijeshrathi@hrc.du.ac.in (B.R.)

**Keywords:** Zika virus, cytotoxicity, 1*H*-1,2,3-triazoles, phthalimide, naphthoquinone, MD simulation

## Abstract

Zika virus (ZIKV) is a mosquito-borne virus belonging to the *Flaviviridae* family and is responsible for an exanthematous disease and severe neurological manifestations, such as microcephaly and Guillain-Barré syndrome. ZIKV has a single strand positive-sense RNA genome that is translated into structural and non-structural (NS) proteins. Although it has become endemic in most parts of the tropical world, Zika still does not have a specific treatment. Thus, in this work we evaluate the cytotoxicity and antiviral activities of 14 hybrid compounds formed by 1*H*-1,2,3-triazole, naphthoquinone and phthalimide groups. Most compounds showed low cytotoxicity to epithelial cells, specially the **3b** compound. After screening with all compounds, **4b** was the most active against ZIKV in the post-infection test, obtaining a 50% inhibition concentration (IC_50_) of 146.0 µM and SI of 2.3. There were no significant results for the pre-treatment test. According to the molecular docking compound, **4b** was suggested with significant binding affinity for the NS5 RdRp protein target, which was further corroborated by molecular dynamic simulation studies.

## 1. Introduction

Zika virus (ZIKV) is a *Flavivirus* that belongs to the *Flaviviridae* family. This epidemic arbovirus is transmitted mainly by the bites of infected mosquitos from the genus *Aedes* and, possibly, *Culex* [1]. The virus has icosahedral symmetry and its genome consists of a single strand of positive-sense RNA, which is translated into structural (C, prM, E) and non-structural (NS1, NS2A, NS2B, NS3, NS4A, NS4B, NS5) proteins [2]. ZIKV was first identified in 1947 when scientists discovered the then unknown virus from a febrile rhesus monkey used in yellow fever research in the Zika forest, Uganda [3]. After 60 years without major outbreaks, ZIKV was responsible for an epidemic in the Yap Islands in 2007 [4] and then in French Polynesia in 2012–2013 [5]. At the beginning of 2015, the first cases in Brazil were reported and the virus caused a serious epidemic resulting in over 200 thousand cases [6]. Several attributes contributed to ZIKV becoming endemic in Brazil, including the abundance of the mosquito vector, the deficiency of basic sanitation, inadequate housing conditions and the deficient public health policies to combat the virus [7].

The most common symptoms developed by patients infected with ZIKV are skin rash, low-grade fever, arthralgia, conjunctivitis, and headache. However, neurological disease of varying degrees of severity have also been reported, including some lethal cases [8,9]. One important neurological disorder associated with ZIKV infection is the Guillain-Barré syndrome, which affects mainly adults and is characterized by the destruction of the myelin sheath of neuronal cells [8]. Importantly, cases of microcephaly and other congenial defects, referred to as congenital Zika syndrome (CZS), have been categorically attributed to ZIKV infection [10]. Despite the severity of associated diseases and its endemicity in tropical areas of the globe, there are neither vaccines nor specific treatment to halt virus replication and alleviate patients’ symptoms. Thus, the development of effective antivirals and vaccines alongside mosquito control strategies are extremely important for combating this important pathogen and preventing new epidemics.

Given their structural diversity, a wide range of biological activities have been explored for the development of therapies. These properties make it possible for these molecules to interact with protein sites and trigger actions at a cellular level. Further structural modification of these compounds may improve the performance of a specific biological activity [11]. Recent studies reported the action of some heterocycles against the Zika virus such as the already known chloroquine [12], sofosbuvir [13], ribavirin, and faviparivir [14], as well as new chemical entities, for example pyrrolo[2,3-d]pyrimidines [15]. A tremendous effort has been and is being invested in the development of effective molecules against ZIKV; however, there no approved anti-ZIKV drugs [16].

Nucleoside analogs are another class of antivirals that have been tested extensively against ZIKV due to their proven role in preventing the replication of RNA viruses. For instance, our research group has identified the thiopurine nucleoside analogue 6-methylmercaptopurine riboside (6MMPr) as a potent inhibitor of ZIKV replication in cell culture [17]. To rationalize our strategy, we consider that phthalimide group is known for its anti-inflammatory activity, an important property in infection events. 1,4-Naphthoquinones have relevant cytotoxic activities. These two parts can be conjugated via click chemistry from organic azides and terminal alkynes generating a variety of 1*H*-1,2,3-triazole compounds (Figure 1).

These three classes of compounds, namely 1,4-naphthoquinone, phthalimide and 1,2,3-triazole inspired our research. These moieties have biological activities against several infections [18,19,20], including ZIKV [21,22]. Our group has synthesized several heterocycle derivatives and tested their activities against human cancer cell lines [23,24,25]. The compounds **1**–**7** (see Figure 2), that contain these cited heterocycles, and will be evaluated, were previously synthesized by our research group, except for compound **5a** [25,26,27]. Here, we expand our initial efforts and evaluate their cytotoxicity to epithelial cells and their antiviral activity against ZIKV. 

## 2. Results

### 2.1. Synthesis of the Compounds ***1**–**7***

Compounds **1**–**7** (Figure 2 and Figure 3, Table 1) were prepared according to the methodologies described previously by our research group [25,26,27]. Only the compound, **5a,** is a new compound. All compounds were characterized through infrared (IR) and nuclear magnetic resonance (NMR) analyses and yields between 62–95% were obtained. For a better discussion of the results, the compounds were classified according to triazole-linked groups.

### 2.2. Cytotoxicity of Triazole Derivatives in Vero Cells

The cytotoxicity of compounds **1**–**7** to Vero cells, an epithelial cell line derived from the kidney of an African green monkey, was tested using the 3-(4,5-dimethylthiazol-2-yl)-2,5-diphenyltetrazolium bromide (MTT) method. Statistical analysis found 20% cytotoxicity concentration (CC_20_) values between 8.87–527.20 µM and 50% cytotoxicity concentration (CC_50_) values from 38.01 to 1189 µM for the tested compounds (Table 2). Based on CC_20_ values, which was the maximum concentration used for antiviral screening, **3b** compound was the least toxic with a value of 527.2 µM. Compounds **2a** and **2b** were the most cytotoxic with values of 8.87 µM and 9.89 µM, respectively. Toxic effects of triazole derivates showed dependence with structural modification. Treatment of cells with compounds linked to naphthoquinone **5a**,**b** and **6a**–**d** resulted in low cell viability, whose CC_20_ ranged between 21.18–74.85 µM. The results from compounds **3a**, **3b** and **4a**, **4b** ranged from 40.69 to 527.20 µM. Compounds **1** (phthalimide-triazole-alcohol) and **7** (sugar-triazole-amino-naphthoquinone) showed median results of 177.10 µM and 119.10 µM, respectively. These results show that the presence of only naphthoquinone in the molecule reduces the values of cytotoxic concentration. The presence of phthalimide group or sugar increases the values of CC_20_ and CC_50_ (Table 2 and Appendix A). 

The cytotoxicity of phthalimide-amino-naphthoquinone-triazoles **3a**, **3b** and **4a**, 4**b** showed dependence with the size of the alkyl chain and presence/absence of the bromine at the naphthoquinone moiety. 

For instance, the compounds **3a** (short alkyl chain, CC_20_ = 55.28 µM) and **4a** (long alkyl chain and bromine, CC_20_ = 40.69 µM) were most cytotoxic than compound **3b** (long alkyl chain, CC_20_ = 527.20 µM) and **4b** (short alkyl chain and bromine, CC_20_ = 136.90 µM). With these data, we observed that the size of the chain and the presence of the bromine atom are decisive for the CC_20_ results. Perhaps the access of the compounds to the cell change accord to these structural differences and chemical properties, as with hydrogen-bond interaction.

In compound **7**, the phthalimide group was exchanged for glucose from compound **4b** and no major changes were found in CC_20_ (119.00 µM).

In general, the presence of phthalimide and/or amino-naphthoquinone groups made the molecules less cytotoxic, to mention some comparisons: (i) **1** (177.10 µM) and **2a**, 2**b** (8.87–9.89 µM): 2-triazole-naphthoquinone is more toxic than phthalimide; (ii) **2a**, **2b** and **6a**–**d** (21.18–39.92 µM), and **5a**, **5b** (74.85–66.93 µM): 2-amino-alkyl-naphthoquinone is less toxic than 2-triazole-naphthoquinone; (iii) **5a**, **5b** with **3b**,**4b** (527.20–136.90 µM): the presence of phthalimide group decrease the cytotoxicity.

### 2.3. Screening of Triazole Derivates

The antiviral activities of the compounds were evaluated by infecting Vero cells with ZIKV and then treating them with the testing molecules. Virus titer in the supernatants of treated wells and its comparison with the negative (cell + virus) and the positive 6-methylmercaptopurine riboside (6MMPr) controls allowed for the determination of the antiviral effect. Only compound **4b** was found to result in a statistically significant reduction in viral titer (Figure 4).

According to the post-infection test performed, compound **4b** proved to be effective in inhibiting the virus at the lowest concentration used. The % viral inhibition showed in Table 3, demonstrates that at 17.11 µM (96.8%) **4b** was most effective than 6MMPr at 7.56 µM (81.1%). Then, using less compound **4b** yielded significant results in inhibiting ZIKV.

### 2.4. Pre-Treatment with Compound ***4b***

Since compound **4b** was the only one in the series with activity against ZIKV, further experiments were conducted to elucidate its mechanism of action in cells that were pre-treated. To this, cells were pre-treated with varying concentrations of compound **4b**, washed, and then infected with ZIKV. The concentrations used were chosen with the aim to preserve cellular viability and, thus, the maximum concentration was the previously determined CC_20_ value (136.90, 68.45, 34.22 and 17.11 µM). There was no significant change in virus titers when compared with negative control (cells + virus, untreated) (Figure 5), suggesting that compound **4b** cannot prevent the entry of ZIKV into Vero cells.

### 2.5. Post-Infection Studies of Compound ***4b***

We then investigated the activity of compound **4b** after ZIKV infection in cells. As already noted in the screening test, **4b** reduced ZIKV growth to levels similar to 6MMPr, a positive control previously described by our group [17]. As in the pre-treatment test, four different concentrations were used, with the maximum being CC_20_ (136.90, 68.45, 34.22 and 17.11 µM). According to the statistical analysis (Figure 5), **4b** inhibits the growth of the virus at all concentrations used, especially in the highest one. It is worth highlighting the constant decrease in the viral titer, even in lower concentrations, while the positive control 6MMPr reduces it only in the highest concentrations. The **4b** half maximal inhibitory concentration (IC_50_) and selectivity index (SI) was 146.0 µM and 2.3, respectively (Table 4).

The treatment time of the compounds for three days post-infection (d.p.i.) and five days post-infection (Figure 6) was also assessed; 6MMPr prevents the growth of the virus in both cases, but with a noticeable decrease if the treatment is longer. Compound **4b** continues to behave similarly in both situations, decreasing the viral titer at all concentrations and at both times of treatment, suggesting that its antiviral activity lasts longer than the reference drug used in our tests (see Appendix A). Statistical analysis compared the concentrations of each compound at each collection time (3 d.p.i. and 5 d.p.i.) with a *t*-test. 

### 2.6. Molecular Docking

After discarding the hypothesis that **4b** acts via a viral entry process and confirming instead that the viral inhibition of compound **4b** occurs in some stage of Zika replication, an investigation of the possible binding site was conducted. The compound, 6MMPr, used as a reference drug, is a riboside analogue, so it was speculated its action would interfere with the RNA processing machinery of the virus. The two viral proteins that bind to RNA are NSP3 helicase and the NS5 RdRp proteins; PDB ID: 6MH3 and PDB ID: 5U04, respectively, were selected to predict the potential target of hit compound **4b** through extensive in-silico studies. 

The RdRp protein has several missing residues on many locations as depicted in Appendix A. These missing residues were filled in the form on loops by Modeller. Among the 9 models, a top model (model 9) was selected based on the ERRAT score, PROSA, Verify 3D, and Procheck (Appendix A, entry 9). The ERRAT score, PROSA, and Verify 3D score of selected models were measured as −70.5882, −7.69, and 84.95, respectively. The model 4 with an ERRAT score of 70.7612 was excluded because of poor <80% Verify 3D score (Appendix A, entry 4). The selected model has only 1.1% residues in a disallowed region (Appendix A, entry 9), which included only 6 residues, Asp346, Ser406, Glu415, Ala537, Arg601, and Lys688, as depicted in Appendix A. Compound **4b** in complex with the helicase (6MH3) protein showed docking score, XP Gscore, and binding free energy values of −3.481 kcal/mol, −3.481 kcal/mol and −47.11 kcal/mol, respectively. On the other hand, compound **4b** complexed with the RdRp (5U04) protein displayed a docking score, XP Gscore, and binding free energy of −5.348 kcal/mol, −5.348 kcal/mol and −52.91 kcal/mol, respectively. In **4b**-helicase complex, compound **4b** interacted to the binding site residues through H-bond (Asn463, Arg462), halogen bond (Trp403, Lys419), salt bridge interaction (Lys358), and pi-cation interaction (Lys419), as represented in Figure 7A. Likewise, in **4b**-RdRp complex, compound **4b** revealed interactions through H-bond (Asp665, Ile799, Arg731), salt bridge interaction (Arg739), halogen bond (Arg731), and pi-pi interaction (Tyr609, Trp797), which are presented in Figure 7B. Molecular docking results suggested that compound **4b** possesses strong binding affinity towards RdRp protein over the helicase.

Next, MD simulation for 100 ns was performed to analyse the stability of the compound **4b** within the binding pocket of both proteins, helicase and RdRp. In **4b**-helicase complex, protein Cα-RMSD reached stability within 10 ns and the results are showcased in Figure 7. The RMSD of compound **4b** achieved stability nearby 28 ns and sustained up to 70 ns. Afterwards, RMSD of **4b** became unstable and escaped from the binding pocket as indicated by the remarkably high RMSD and trajectory analysis (Figure 8). The ligand RMSF showed that its each atom fluctuated more than 6Å, as presented in Appendix A. The interaction with water may be responsible for the instability of compound **4b,** as displayed in Appendix A.

On other hand, protein Cα-RMSD of **4b**-RdRp complex, stabilized within 10 ns and remained stable throughout the simulation period (100 ns) as presented in Figure 8. Compound **4b** RMSD suggested the structural and conformational stability, where fluctuation was observed within an acceptable range, 3 Å (Figure 9A). Each atom of the compound **4b** fluctuated within 3.5 Å (Figure 9B). Compound **4b** showed interactions with residues Tyr609, Trp797, Arg731, and Ile799 and thus supported docking results (Figure 9C). The Ramachandran mapping of compound **4b**-RdRp complex after MD simulation showed only 0.6% (Asn454, Glu695, and Trp748) residues in the disallowed region for the protein RdRp, which indicated good stereo-chemical geometry of the residues, as presented in Appendix A.

## 3. Discussion

The search for effective treatments to combat symptoms and protect against ZIKV remains in progress, since there are no options available in the market. This work explored the potential of synthetic compounds in the search of viable treatment options against this devasting virus.

According to the structural analysis of the compounds, some aspects related to the cytotoxic effect on Vero cells were highlighted. The compounds **2a** and **2b** (naphthoquinone-triazole-alcohol), whose cell viability results were more cytotoxic based on their CC_20_ value, presents the 1,4-naphthoquinone directly linked with 1*H*-1,2,3-triazole nucleus; on the other hand, in general, 2-amino-1,4-naphthoquinone decrease the toxicity level [27]. We believe that some conformational factor hydrophobic effects are present. Michael acceptor group (**3a**/**3b**), and “halogen bond”-bromide- (**4a**/**4b**) can help with protein-ligand complex interactions. All these factors can promote different results in cytotoxicity. The **4a** cytotoxicity is likely due to this compound being more lipophilic (logP = 3.23, Table 2) and may accumulate in the cell membrane with toxic consequences. This effect is shown in CC50 (**4a** = 68.78 µM). The phthalimide group, represented by 4-phenyl-1-[2-(phthalimido-2-yl)ethyl]-1-*H*-1,2,3-triazole, showed low cytotoxicity for macrophages (peritoneal and J774A.1) and fibroblast cells [28]. 

Naphthoquinone-triazoles **6a**–**d** linked to carbohydrates decreased the cytotoxicity, i.e., at 50 µM they maintained cell viability around 80%, except for **6b**. The amino-naphthoquinone-triazoles **5a** and **5b** behaved contrarily compared to **2a**, **2b** and **6a**–**d**, being less cytotoxic, with cells remaining viable until the concentration of 400 µM. 

Naphthoquinone compounds have a variety of activities described in the literature, including cytotoxicity in Vero cells. Gonzaga et al. (2019) used the MTT method to test the cytotoxicity of compounds derived from bis-naphthoquinones [21]. In general, the presence of the group, even in duplicate, does not increase cytotoxicity to Vero cells.

The screening reveals little or no action of the compounds **1**–**7** against ZIKV, except for **4b**, which was the only one with statistically significant results when compared to the non-treated control. Pre-treatment and post-infection tests were performed with **4b** to elucidate the possible mechanism of action and determine the IC_50_ and SI of values, 146.0 µM and 2.3, respectively. The results indicate that the compound **4b** probably acts in some stage of the virus replication after it has entered the cell. Overall, compound **4b** exhibited significant anti-ZIKV activity without any apparent cytotoxicity. 

Lima et al. (2019) tested triazole compounds against ZIKV with Vero cells. The method used was MTT in both analyses, thus verifying the viability of the cells in the tests. Lima’s triazole series showed high cytotoxicity. The antiviral activity were the best ZIKV inhibitors, considering at least 50% of cell viability [22]. 

The analysis with the compound-based 1*H*-1,2,3-triazoles strengthens the principle that the presence of nitrogen groups increases the biological activity. The structural modification of the tested groups driven by in-silico techniques helped us to understand the negative response of the tested compounds against ZIKV, as well as the improvement of the **4b** response in viral inhibition.

Concerning the docking score, XP Gscore and binding free energy, compound **4b** was suggested with a significant binding affinity towards RdRp protein when compared with helicase. These results were further corroborated by MD simulation studies. However, validation experiments are necessary before concluding the possible target for the hit compound **4b**.

## 4. Materials and Methods

### 4.1. Chemistry

Drugs: the compounds **1**–**7** (Figure 1) were synthesized by our research group. The formation of the triazole derivatives was carried out by copper(I)-catalysed azide-alkyne cycloaddition (CuAAC) through the reaction between phthalimide-azide or naphthoquinone-azide and 2,3-unsaturated glycosides or alkynyl alcohols. The synthesis of 2,3-unsaturated glycosides **6a**–**d** was carried out using the montmorillonite K-10/iron(III) chloride hexahydrate catalysis described by Melo et al. (2015) [25]. 

The new compound **5a** was synthesized using Method E (Table 1) [26]. The ^1^H and ^13^C NMR are showed in the Appendix A. Characterization of 2-[2-(4-(3-Hydroxypropyl)-1H-1,2,3-triazol-1-yl)]-ethylamino-1,4-naphthoquinone (**5a**): Yield = 90%; red solid; Mp 143–145 °C. ^1^H NMR (400 MHz, DMSO-*d_6_*): δ 7.98–7.92 (m, 2H, H_Naph_), 7.87 (s, 1H, H_Triaz_), 7.84–7.72 (m, 2H, H_Naph_), 7.49 (br s, 1H, NH), 5.70 (s, 1H, H-3_Naph_), 4.54 (t, 2H, *J* = 6.0 Hz, CH_2_N_Triaz_), 4.39 (t, 1H, *J* = 5.0 Hz, OH), 3.66–3.64 (m, 2H, CH_2_NH), 3.40 (m, 2H, CH_2_OH), 2.60 (m, 2H, CH_2_-C4_Triaz_), 1.70 (m, 2H, CH_2_). ^13^C NMR (100 MHz, DMSO-*d_6_*): δ 202.2, 182.1, 148.7, 145.0, 135.3, 134.8, 132.7, 130.8, 126.3, 125.8, 122.8, 100.5, 60.5, 47.7, 42.4, 32.7, 22.1. Anal. calcd. for C_17_H_18_N_4_O_3_: C, 62.57; H, 5.56; found: C, 62.71; H, 5.49. 

After molecular structure characterization by infrared (IR), and ^1^H and ^13^C NMR, the compounds were tested for ZIKV infection. The stock solutions were made at a concentration of 200 mM with DMSO (Sigma-Aldric, San Luis, MO, USA). The compounds dilutions were made from the stock solution with Dulbecco’s Modified Eagle Medium (DMEM) (Thermo Fisher Scientific, Waltham, MA, USA) reaching the maximum concentration of 0.8% DMSO.

A stock solution of the riboside nucleotide analog 6MMPr (Sigma-Aldrich, Louis, MO, USA) of 1 mM was made in Milli-Q water and the compound was used as a positive control in antiviral tests. Dilutions of 60.50, 30.25, 15.13 and 7.56 µM were used according to the cytotoxicity methodology and CC_20_ values used by de Carvalho and co-workers [17].

### 4.2. Cells and Viruses

Vero cells (CCL-81) were used in all phases of the in vitro testing. The cultivation was performed with products from the company, Thermo Fisher Scientific. DMEM supplemented with 10% foetal bovine serum (FBS) and 1% penicillin/streptomycin and incubated at 37 °C and 5% CO_2_ atmosphere on incubator Heracell VIOS 250i (Thermo Fisher Scientific).

A strain of ZIKV/H.sapiens/Brazil/PE243/2015 (abbreviated to ZIKV PE243; GenBank accession number KX197192.1) was used in antiviral tests. The viral stock was made in Vero cells and grown at 37 °C and 5% CO_2_ atmosphere. The endpoint dilution assay was the method used to obtain the TCID_50_/mL titer of this stock.

### 4.3. Titration of Stock and Samples Virus

96-well plates with 1 × 10^4^ cell/well were used in this assay and prepared 24 h previously. Seven serial dilutions of the virus were prepared in DMEM + 2% FBS culture medium. After removing the culture medium from the plate, 50 µL of each dilution was added to the wells in several repetitions. The plate was incubated at 37 °C and 5% atm CO_2_ for 1 h. After this interval, each well of the plate received an additional 100 µL of DMEM + 2% FBS culture medium. The plate was incubated again, in the same parameters, for 5 days. Some columns of the plate were selected for use as a cell control (culture medium only, no virus). The reading was performed in an inverted optical microscope AE2000, Motic (Motic, Kowloon Bay, Hong Kong) counting positives wells by the presence of cytopathic effect and the determination of the sample title was performed with the aid of the Microsoft Office Excel program (Microsoft^®^ Office, Redmond, WA, USA).

### 4.4. Cytotoxicity Test with MTT Method

96-well plates with 1 × 10^4^ cell/well were used in this assay and prepared 24 h previously. The compounds were diluted in a 2:1 ratio so that a maximum of 0.8% DMSO was used. The concentrations used were 800, 400, 200, 100, 50, 25 and 12.5 µM. Dilutions were added to the wells in triplicate and incubated at 37 °C and 5% CO_2_ atmosphere. After 5 days, the wells were emptied and a 50 µL of 1 mg/mL MTT (3-(4,5-dimethylthiazol-2-yl)-2,5-diphenyltetrazolium bromide) (Sigma-Aldrich) solution was added, and the cells were incubated for 3–4 h for formation of formazan crystals through the mitochondria of viable cells. Then, medium was removed and 100 µL DMSO was added to the wells to solubilize the formazan crystals. With the aid of an incubator Tecnal TE-420 (Tecnal Equipamentos Científicos, Piracicaba, São Paulo, Brazil), the plates were homogenized at 120 RPM at 37 °C and the optical densities were read at 570 nm on a spectrophotometer BioTekTM ELx800TM (BioTek Instruments, Winooski, VT, USA). The statistical analysis of these data revealed the values of CC_20_ and CC_50_, both in µM, which was used in the screening test.

### 4.5. Screening Test

The screening of the compounds was carried out with the CC_20_ value (µM) to determine the most viable compounds for more detailed antiviral tests. 48-well plates with 2.5 × 10^4^ cells/well were prepared with a previous 24 h. The culture medium from the plate was removed and the wells were inoculated with a viral solution at MOI 0.1, except cell control (which received only culture medium + 2% FBS). During 2 h of incubation at 37 °C and 5% CO_2_ atmosphere, the solutions with the compounds are prepared with DMEM + 2% FBS. After incubation, the medium was discarded, and the wells were washed with 1X phosphate buffer solution (PBS). The solutions with the compounds were added to the wells, in triplicate, and then incubated for 5 days under the same conditions. After incubation, the supernatant from each well was collected and stored at −80 °C. The samples were then titrated and analysed statistically.

### 4.6. Pre-Treatment Activity

To prepare, 48-well plates with 2.5 × 10^4^ cells/well were prepared 24 h previously. Compound **4b** was diluted in a 2:1 ratio in culture medium with concentrations of 136.90 µM (CC_20_), 68.45 µM (CC_20_/2) 34.22 µM (CC_20_/4) and 17.11 µM (CC_20_/8). The dilutions were inoculated into the wells in triplicate and the cells incubated at 37 °C and 5% CO_2_ atmosphere. After 1 h, the culture medium with the compounds was removed and the wells were washed with 1X PBS. A ZIKV solution at MOI 0.1 was inoculated into the treated and control wells. The cells were again incubated for 5 days. The supernatants from the wells were collected and titrated.

### 4.7. Post-Infection Activity

48-well plates with 2.5 × 10^4^ cells/well were prepared 24 h previously. A ZIKV solution at MOI 0.1 was inoculated into the wells and the cells incubated at 37 °C and 5% CO_2_ atmosphere. After 2 h, the culture medium was removed, and the wells were washed with 1X PBS. 2:1 dilution of **4b** (136.90 µM, 68.45 µM, 34.22 µM and 17.11 µM) and 6MMPr (60.50 µM, 30.25 µM, 15.13 µM and 7.56 µM) were added in triplicate. The cells were again incubated for 72 h and 120 h. The supernatants from the wells were collected and titrated.

### 4.8. Molecular Docking

The coordinate structure of NSP3 (PDB: 6MH3) and NSP5 (PDB: 5U04) proteins were collected from RCSB website (https://www.rcsb.org, accessed on 10 February 2021). The NSP3 and NSP5 also known as helicase and RNA dependent RNA polymerase (RdRp) protein, respectively. The RdRp protein had several missing residues (Appendix A), so missing loops were filled with the help of modeller software [29]. The best model was selected among 15 models based on ERRAT [30], PROSA [31], Verify3D [32], and PROCHECK [33] (Appendix A). The computational work performed using Schrodinger software (Schrodinger release 2020-1 license dated 20 November 2020). Both protein structures were prepared to minimize the structural defects by using protein preparation wizard before docking studies [34,35,36]. Before docking compound **4b** structure was also prepared by the Ligprep [35,37]. The Epik module of Schrodinger was used to predict the ionization states of compound **4b** at pH 7 ± 2 as well as tautomers generated [38]. In-silico study was carried out under OPLS2005 forcefield.

#### 4.8.1. Molecular Docking of Designed Chemical Library 

The binding site of both the proteins was predicted by sitemap [39]. The best site was selected based on site score (Appendix A). The co-ordinate of binding site for RdRp protein includes 26.17, 66.84, and 103.42, while the helicase protein includes −5.54, 2.84, and −6.55. Compound **4b** was docked at XP precision to both the proteins in a site-specific manner using Glide module of Schrödinger suite [40]. The Van der Waals radii scaling factor and partial charge cut-off was 0.8 and 0.15, employed for docking study, respectively. The binding free energy for both the complexes were also calculated by prime MMGBSA [41].

#### 4.8.2. Molecular Dynamics (MD) Simulation 

To validate the docking results, both the protein complexes of compound **4b** with helicase and RdRp protein were selected for 50 ns MD simulation [42]. These complexes were solvated in TIP3P [43] water model and 0.15 M NaCl to mimic a physiological ionic concentration. The stereo-chemical geometry of 5HGL protein residues was measured by Ramachandran map by procheck [33].

### 4.9. Statistical Analysis

After reading the optical densities of the cytotoxicity test revealed by the MTT method, the calculation of the cell viability percentage was performed in the Microsoft Office Excel 15 (Microsoft^®^ Office) program using the following formula:% Viability = OD _sample_ × 100/OD_cellular control_(1)

The determination of the CC_20_ and CC_50_ values was obtained through the linear regression analysis of the XY graph generated by the values of the concentrations used in the test and the triplicate values of the viability. In the post-infection and pre-treatment tests, ANOVA and Dunnett tests were used, considering *p* > 0.05 as the minimum significance value. All tests were performed in GraphPad Prism v.6.0 program (GraphPad Software, Inc., San Diego, CA, USA).

## 5. Conclusions

Heterocyclic compounds have several activities recorded in the literature. The triazole and naphthoquinone groups have already been reported with antiviral activity against the Zika virus, while the phthalimide group has no previously reported activity against the virus. Fourteen compounds were tested in this study. In the cytotoxicity test, the CC_20_ (8.87–527.20 µM) and CC_50_ (38.01–1189.00 µM) values were quite diverse, showing high to low values, but all of them were feasible for the antiviral screening. Compound **3b** stood out with the highest values of CC_20_ and CC_50_. The screening revealed compound **4b** as the most active against ZIKV, and the results confirm a post-infectious antiviral activity. Docking results coupled with score, XP Gscore and binding free energy suggested compound **4b** with strong binding affinity for NS5 RdRp protein target of ZIKV, which was further supported by MD simulations. However, validation experiments are necessary before concluding the possible target for the hit compound **4b**, which could be a part of our future investigations. Lead optimization, in conjunction with in-silico analysis, could deliver new versions of the molecule with significantly increased activity against ZIKV. In addition, in vivo tests will be important to confirm the efficacy of these promising molecules in adequate ZIKV animal models.

## Figures and Tables

**Figure 1 molecules-26-05869-f001:**
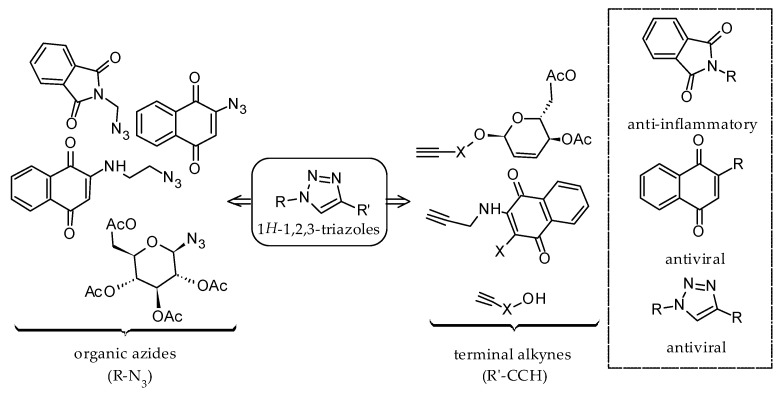
Strategy for molecular diversity towards antiviral 1*H*-1,2,3-triazoles.

**Figure 2 molecules-26-05869-f002:**
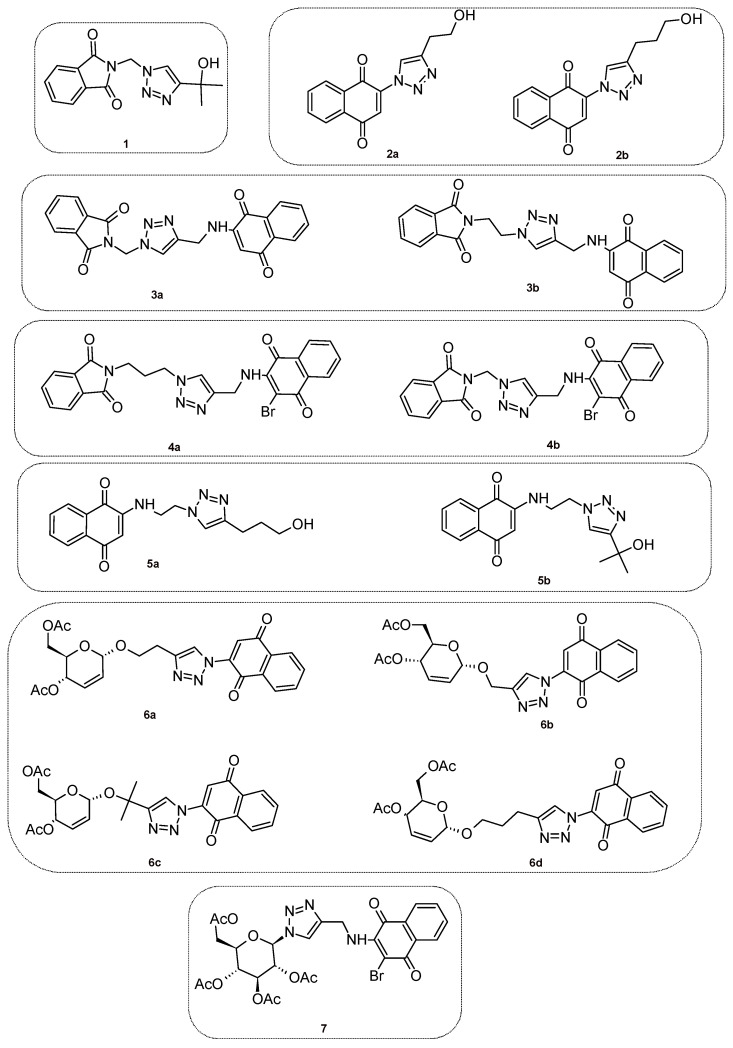
1*H*-1,2,3-Triazole derivatives **1**–**7** tested against ZIKV.

**Figure 3 molecules-26-05869-f003:**
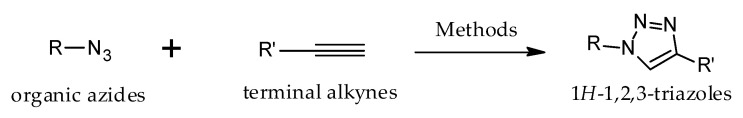
Synthesis of 1*H*-1,2,3-triazoles from organic azides and terminal alkynes.

**Figure 4 molecules-26-05869-f004:**
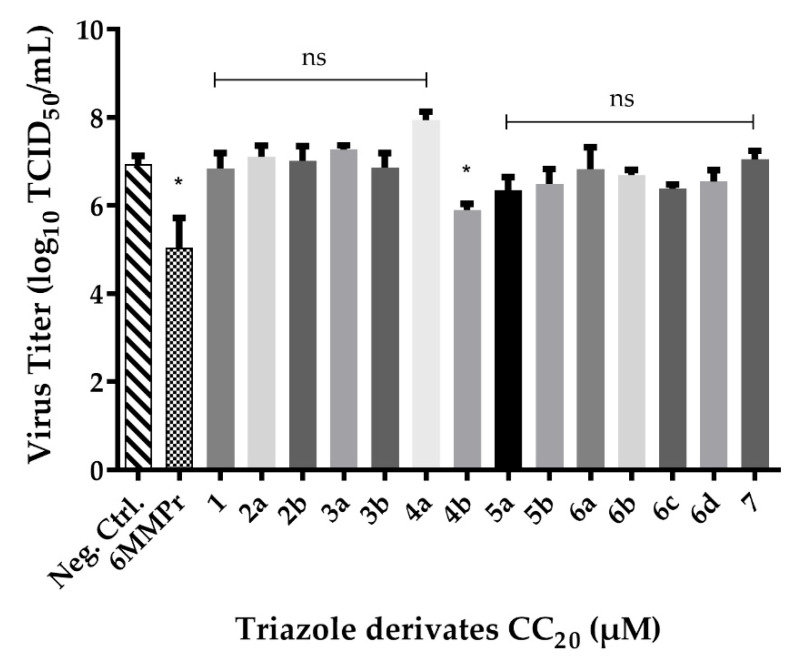
Graphic result of all triazole derivates in the screening test for anti-ZIKV activity. The screening test was made for post-infection with ZIKV MOI 0.1 and treated with CC_20_ values from heterocyclic compounds discovered in the cytotoxicity test. After five days, the supernatant was collected and titrated by TCID_50_ method. The experiment was conducted in triplicates and values are the mean and the standard deviation (SD). In the GraphPad Prism program, Dunnett’s ANOVA analysis was used to determinate the best compounds comparing with control. * = significative; ns = not statistically significant for *p* < 0.05.

**Figure 5 molecules-26-05869-f005:**
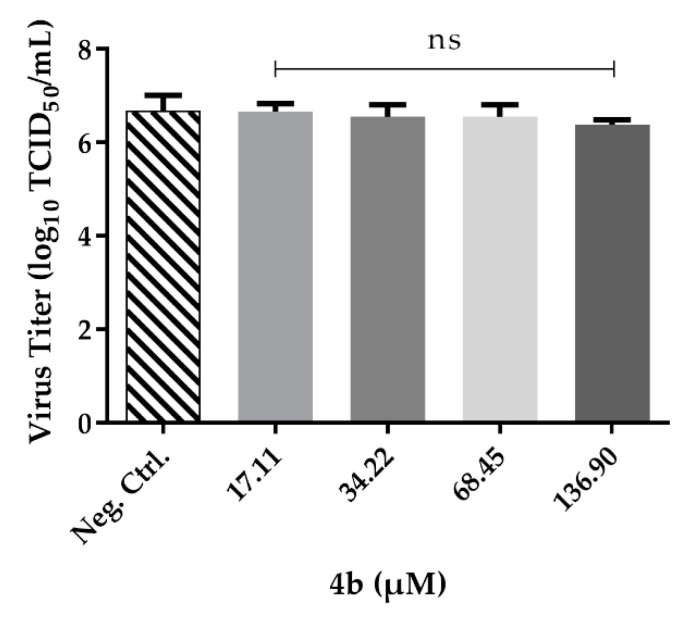
Graphic result for compound **4b** in the pre-treatment test to ZIKV. Vero cells in a 48-well cell plate were treated with **4b** in different concentrations and after 1 h the supernatant was discarded. Cells were washed with PBS 1X, ZIKV MOI 0.1 was inoculated, and the plate was incubated again. After five days, the supernatant was collected and titrated by TCID_50_ method. The experiment was conducted in triplicates and values are the mean and the SD. In the GraphPad Prism program, Dunnett’s ANOVA analysis was used to determinate the best concentrations comparing with control. ns = not statistically significant for *p* < 0.05.

**Figure 6 molecules-26-05869-f006:**
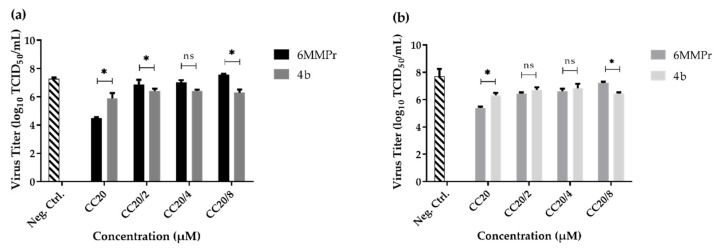
Comparison of anti-ZIKV activity between 6MMPr and **4b** after three- and five-days post-infection (d.p.i.). 6MMPr and compound **4b** were submitted to a post-infection test when Vero cells was infected with ZIKV at MOI 0.1 for 2 h then treated with compounds. Concentrations used in the test corresponded to CC_20_ (**4b** = 136.90 µM; 6MMPr = 60.50 µM), CC_20_/2 (**4b** = 68.45 µM; 6MMPr = 30.25 µM), CC_20_/4 (**4b** = 34.22 µM; 6MMPr = 15.13 µM) and CC_20_/8 (**4b** = 17.11 µM; 6MMPr = 7.56 µM). The experiment was done in triplicates and values are the mean and the SD. Statistical analysis was made between compounds using *t*-test on GraphPad Prism program. * = significative; ns = not statistically significant for *p* < 0.05. (**a**) 3 d.p.i. from both compounds. (**b**) 5 d.p.i. from both compounds.

**Figure 7 molecules-26-05869-f007:**
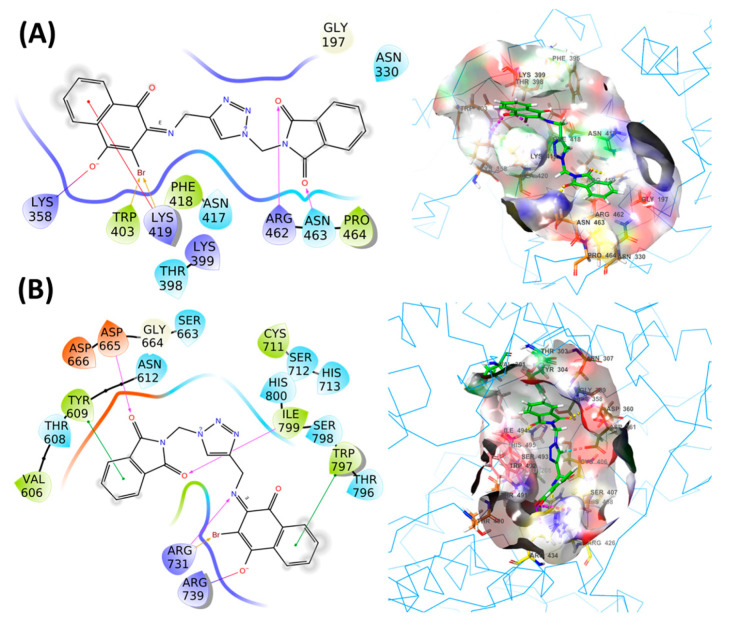
2D-3D diagram of ligand-protein complex. (**A**) **4b**-helicase (6MH3) complex. (**B**) **4b**-RdRp (5U04) complex.

**Figure 8 molecules-26-05869-f008:**
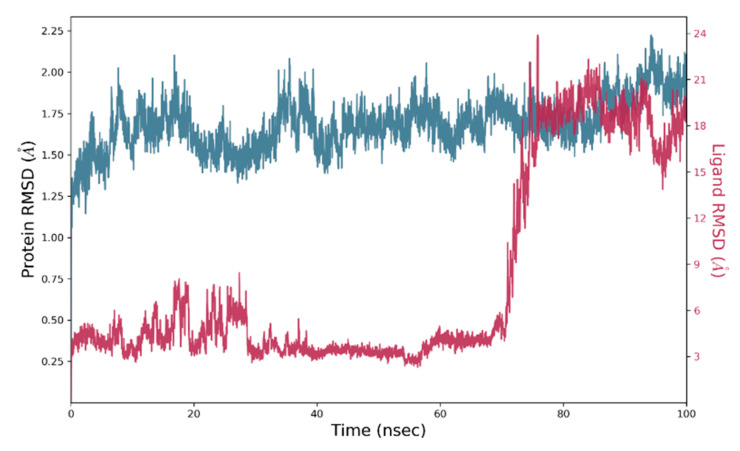
RMSD plot obtained for 4b-helicase (6MH3) complex: protein Cα and compound **4b** RMSD shown in blue and red color, respectively.

**Figure 9 molecules-26-05869-f009:**
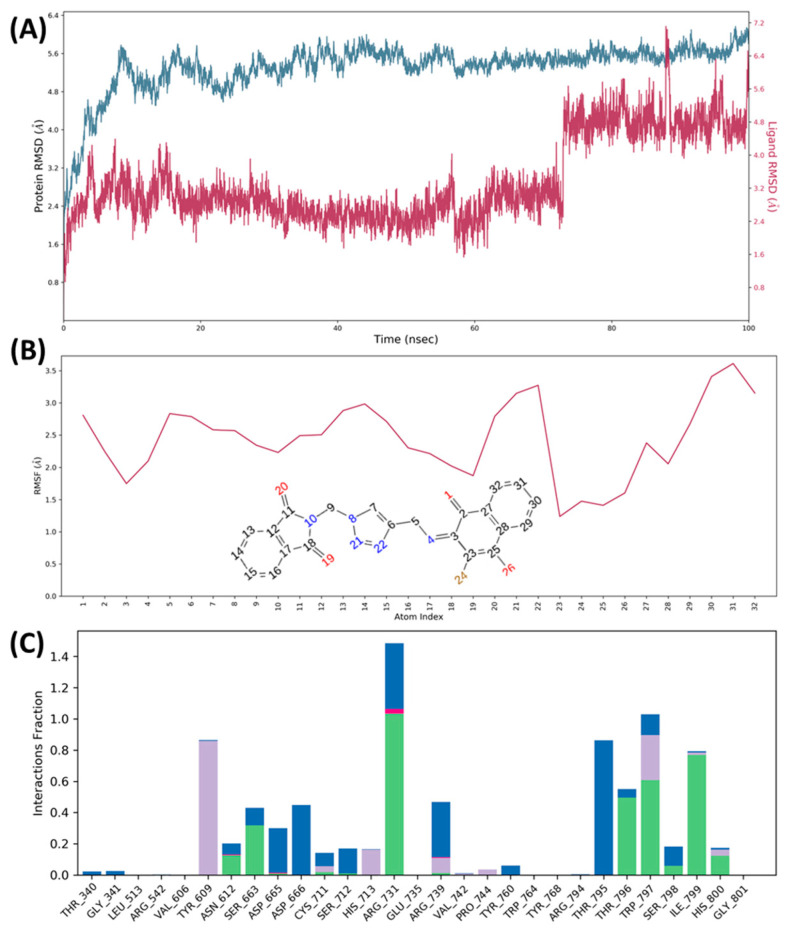
Molecular dynamics of **4b**-RdRp (5U04) complex. (**A**) RMSD plot obtained for protein Cα and compound **4b** shown in blue and red colour, respectively; (**B**) Ligand RMSF-fluctuations of each atom of compound **4b** with respect to the protein; (**C**) Histogram plot showed interacting residues with compound **4b** during MD simulation (H-bond: green, hydrophobic: grey, salt bridge interaction: pink, and water bridge interaction: blue).

**Table 1 molecules-26-05869-t001:** Synthesis of 1*H*-1,2,3-triazole derivatives **1**–**7**.

Entry	Products	Methods	Yields, %	Reference
1	**1**	A	92	[26]
2	**2a,b**	B	85–95	[25]
3	**3a,b**	C	70–95	[26]
4	**4a,b**	D	70–88	[27]
5	**5a**	E	90	New compound
	**5b**	E	86	[26]
6	**6a–d**	F	62–85	[25]
7	**7**	D	85	[27]

Method-A: DMF/CuI//Et_3_N/ultrasound/30 min.; Method-B: CH_3_CN/CuI/0 °C/12 h/absence of light; Method-C: CH_3_CN/CuI/r.t./20 h; Method-D: DMF/CuI/Et_3_N/r.t./60 min; Method-E: DMSO/CuI/Et_3_N/r.t./2 h; **F**: CH_3_CN/CuI/0 °C/12 h.

**Table 2 molecules-26-05869-t002:** Cytotoxicity assay in Vero cells (CC20 and CC50).

Compound	CC_20_ (µM) ^1^	CC_50_ (µM) ^2^	logP ^3^
**1**	177.10	680.10	1.44
**2a**	8.87	38.01	0.99
**2b**	9.89	38.81	1.26
**3a**	55.28	469.80	2.21
**3b**	527.20	1189.00	2.22
**4a**	40.69	68.78	3.23
**4b**	136.90	330.20	2.94
**5a**	74.85	293.90	1.05
**5b**	66.93	298.20	1.38
**6a**	39.92	68.05	2.07
**6b**	21.18	48.69	1.86
**6c**	25.73	58.14	2.67
**6d**	38.61	69.50	2.34
**7**	119.10	334.50	1.80
**6MMPr** ^4^	60.5	291	0.55

^1^ CC_20_ (20% cytotoxic concentration) refers to compound concentration that causes a 20% reduction in cell viability. ^2^ CC_50_ (50% cytotoxic concentration) refers to compound concentration that causes a 50% reduction in cell viability. ^3^ The logP (the logarithm of the octanol-water partition coefficient): was calculated using the Molinspiration molecular property (http://www.molinspiration.com).^4^ See reference [17].

**Table 3 molecules-26-05869-t003:** % Viral Inhibition.

Compound	4b			6MMPr	
Concentration (µM)	3 d.p.i. ^1^ %	5 d.p.i %	Concentration (µM)	3 d.p.i. %	5 d.p.i. %
CC_20_ = 136.90	94.9	97.5	60.50	99.8	99.7
CC_20_/2 = 68.45	85.6	93.7	30.25	54.4	96.8
CC_20_/4 = 34.23	85.8	90.8	15.13	42.7	95.1
CC_20_/8 = 17.11	88.8	96.8	7.6	−93.4 *	81.1

^1^ d.p.i.: days post-infection. * The negative sign refers to increased virus titer compared to control.

**Table 4 molecules-26-05869-t004:** Results for post-infection test with 6MMPr and **4b**.

	IC_50_ ^1^ (µM)	SI ^2^	% Viral Inhibition
**4b**	146.0	2.3	91.1
**6MMPr**	24.5	11.9	97.8

^1^ IC50 (50% inhibition concentration) refers to compound concentration that caused a 50% inhibition in virus replication. ^2^ SI (selectivity index) refers to the ratio between the CC_50_ and IC_50_ values.

## Data Availability

The authors declare no competing financial interest. The study is done on Schrodinger maestro 2020-1 version platform. This software can be downloaded from https://www.schrodinger.com/. Another software, Desmond Release 2020-1 was used for molecular dynamics simulations is freely available and can be downloaded from “https://www.deshawresearch.com/resources_desmond.html”. All the protein structure, ligand-protein complex file, and MD simulation data are made available on Github https://github.com/premprakash11241/Searching-Anti-Zika-Virus-Activity-in-1H-1-2-3-Triazole-Based-Compounds.

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
