# Peer review of "Searching Anti-Zika Virus Activity in 1H-1,2,3-Triazole Based Compounds"

_molecules, 2021, doi:10.3390/molecules26195869_

Round 1

Reviewer 1 Report

In this paper, the optimal compound was selected through experiments, and then molecular dynamics simulation was carried out to prove that compound 4b had certain antiviral activity. But as you said, cellular and in vivo studies are needed to more directly demonstrate the effectiveness of the compound.

As you can see from your experiment, the effect of compounds 4b is almost independent of the concentration of the drug when the compounds is treated for a long time (5 days). So that compounds 4b, as a newly developed antiviral compound, what are the advantages over the currently known antiviral drugs?

Minor suggestion:

Line 159, "cannot established" should be revised.

Author Response

Response to Reviewer 1 Comments

1) In this paper, the optimal compound was selected through experiments, and then molecular dynamics simulation was carried out to prove that compound 4b had certain antiviral activity. But as you said, cellular and in vivo studies are needed to more directly demonstrate the effectiveness of the compound.

Response: I agree with referee; however, even in vivo experiment is not a guarantee for active compounds in vitro, and the opposite is also valid. In fact, despite a modest activity of 4b, these allow future research to modify this framework.

2) As you can see from your experiment, the effect of compounds 4b is almost independent of the concentration of the drug when the compounds is treated for a long time (5 days). So that compounds 4b, as a newly developed antiviral compound, what are the advantages over the currently known antiviral drugs?

Response: Our experiments for 3 days or 5 days were able to show viral inhibition. Despite the limitation, it is known that the viral resistance is a problem, and the new tested compounds can compose a future library to access anti-ZIKV activity. We can note that 17.11 µM still maintains the viral inhibition in 96.8%. The Table 3 was included and comments in the text (lines 178-185).

3) Line 159, "cannot established" should be revised.

Response: It was revised (lines 195-196).

Reviewer 2 Report

In the manuscript "Searching Anti-Zika Virus Activity in 1H-1,2,3-triazole based compounds " sent to me for review the authors describe the results of the research on the cytotoxicity and antiviral activities of 14 compounds formed by a combination of such structural elements as naphthoquinone, 1,2,3-triazole ring, and phthalimide or a sugar moiety.

The work begins with a brief and logical introduction justifying the need to find an effective drug to treat diseases caused by the Zika virus belonging to the Flaviviridae family. The following paragraph mentions the compounds tested against the Zika virus, focusing mainly on nitrogen-containing heterocyclic compounds, including quinoline derivatives, isoflavones and nucleoside derivatives. Given the large number of publications devoted to the study of antiviral activity against the Zika virus of various compounds, including natural ones, this fragment seems a bit too modest. It was enough to read only one publication, which appeared in 2019 in Pharmaceuticals (doi: 10.3390 / ph12030101) to find out the number of compounds tested for anti-Zika activity. The justification for the selection of structural elements introduced into the compounds tested by the authors is presented even more briefly. Neither the role of the 1,2,3-triazole system, which appears even in the title of the paper, nor the one introduced in the 6a-6d series derivatives and the 7 sugar fragment, has been explained. Perhaps the authors described their line of reasoning when designing the studied structures in earlier works, however, the current manuscript is reviewed and it should contain justify the selection of certain structural elements.

Considering that the authors no longer dealt with the synthesis of new compounds, but only used the compounds they had in their resources and which were described in the three earlier papers, examining just 14 derivatives does not seem to be very impressive work. The more so as the tested compounds, maybe apart from the compound 4b, did not exhibit antiviral activity. Even the more widely studied compound 4b did not exhibit high antiviral activity (IC50 of 146 µM) with a low SI of 2.3. In this situation, it was begging to synthesize a few more structurally diverse derivatives and check their activity.

In discussing the results, the authors analyze the influence of individual structural elements on the cytotoxicity demonstrated by the final compounds. It is worth emphasizing that the fundamental relationship that can be noticed is the undeniable influence on the increase in cytotoxicity of the direct connection of the 1,2,3-triazole system with naphthoquinone. The remaining structural elements seem to have a much smaller impact in this situation.

In the conclusions, the authors mention the need for further research, including in vivo research, which "are essential to confirm the effectiveness of the promising compounds in more complex organisms". However, I am afraid that with such a poor activity of the compound 4b, comparing, for example, to the compound 6MMPr used as a reference drug, in vivo tests are unlikely to occur and it is pointless to mention them.

The authors tried to enrich the work by elucidating the mechanism of action of compound 4b and adding the results of molecular docking, however it still seems that this is not enough for the work to be worth publishing in a journal such as Molecules.  

However, if the work will be published, it is also worth correcting linguistic and editorial errors that appear in the manuscript, e.g.

Line 44: the given number of 205,578 cases of should be an integer number. The comma should definitely be omitted.

Line 74: “Another class of antivirals are”- “are” should be replaced with “is”.

Line 111: “These results shows that the presence of only naphthoquinone on the molecule reduce the values of cytotoxic concentration”- “on” should be replaced with “in”, “reduce” should be replaced with “reduces”.

Reviewer 3 Report

All comments to authors are in the attached document.

Round 2

Reviewer 2 Report

I carefully read the authors' responses to my remarks as well as the revised manuscript of the work entitled "Searching Anti-Zika Virus Activity in 1H-1,2,3-Triazole Based Compounds". I am glad that the authors took into account albeit a small part of my comments and suggestions, which allowed for some "enrichment" of the submitted work. Unfortunately, as far as the introduction is concerned, I still consider the discussion of a current state of knowledge section on compounds tested against Zika virus to be too modest. In my review, I mentioned about numerous works devoted to this topic and gave one of them as an example. The authors, however, limited themselves to reading only the paper I have mentioned and stating that "in this article, the compounds are" repurposing drugs". There aren't lead compounds from basic research". Perhaps, in such a situation, a more in-depth review of the literature should have been carried out to see if this fragment of the introduction should be supplemented.

Let me also disagree with the statement "many publications report testing fewer than 10 compounds against a given virus", which appeared in response to my remark about the small number of compounds tested, especially in a situation where they were not new compounds but already described in three other works. Thus, the authors did not have to spend additional time on their preparation. There are indeed such works, but they are not necessarily role models. In my opinion, in science we should try to imitate the best, not the average.

Table 1 added to the work only confirms that the synthetic contribution of the work in this case was insignificant, because the authors had a vast majority of the tested compounds for a long time. However, it can be seen that at least one compound (5a) is new. In turn, inserting Figure 1 allows at least to understand the idea behind the authors while designing the tested connections. The other changes made to the manuscript, however, are rather "cosmetic" changes and do not quite fit the concept of "major revision".

Author Response

Response to Reviewer 2 Comments

I carefully read the authors' responses to my remarks as well as the revised manuscript of the work entitled "Searching Anti-Zika Virus Activity in 1H-1,2,3-Triazole Based Compounds". I am glad that the authors took into account albeit a small part of my comments and suggestions, which allowed for some "enrichment" of the submitted work.

Unfortunately, as far as the introduction is concerned, I still consider the discussion of a current state of knowledge section on compounds tested against Zika virus to be too modest. In my review, I mentioned about numerous works devoted to this topic and gave one of them as an example. The authors, however, limited themselves to reading only the paper I have mentioned and stating that "in this article, the compounds are" repurposing drugs". There aren't lead compounds from basic research".  Perhaps, in such a situation, a more in-depth review of the literature should have been carried out to see if this fragment of the introduction should be supplemented.

Answer: All suggested changes have been made. We have cited #15 and 16.

Let me also disagree with the statement "many publications report testing fewer than 10 compounds against a given virus", which appeared in response to my remark about the small number of compounds tested, especially in a situation where they were not new compounds but already described in three other works. Thus, the authors did not have to spend additional time on their preparation. There are indeed such works, but they are not necessarily role models. In my opinion, in science we should try to imitate the best, not the average.

Answer: We agree with the reviewer that we should always aim high. We have tested 14 compounds and one of them had never been published before. The preparation and synthesis of these compounds in amounts sufficient for the antiviral tests done here required a considerable amount of funds and time at the bench. These are not commercially available compounds that one could just purchase. Thus, this manuscript demanded a significant amount of work. 

Table 1 added to the work only confirms that the synthetic contribution of the work in this case was insignificant, because the authors had a vast majority of the tested compounds for a long time. However, it can be seen that at least one compound (5a) is new. In turn, inserting Figure 1 allows at least to understand the idea behind the authors while designing the tested connections. The other changes made to the manuscript, however, are rather "cosmetic" changes and do not quite fit the concept of "major revision".

Answer: We respect the reviewer´s point of view, but we do believe that our manuscript required a significant body of work, and it provides significant contributions to the field of antivirals discovery. The manuscript has a total of 8 figures and 4 tables; thus we strongly believe that it has an adequate amount of information required for a strong scientific paper.

Reviewer 3 Report

1) In lines 60 to 78, the authors cite several works described in the literature that do not justify the importance of choosing naphthoquinone, phtalimide and 1,2,3-triazole nuclei as pharmacophores in planning new hybrids.

Response: We presented a general description of heterocyclic compounds with anti-ZIKV activity. There are a limited anti-ZIKV activity reports in the literature from triazole.

Response to authors: The paragraph in lines 60-78 should be redone. It has no connection with the posterior paragraph. Please keep the explanation about the use of nucleosides and remove all other examples that are unrelated to the work. Cite the limited anti-ZIKV activity reports in the literature about 1,2,3-triazole and move the lines 82 to 86 to this paragraph. These lines explain about the choice of 1,4-naphthoquinone and phthalimide in the research.

2) Why non-structural proteins NSP3 helicase and NSP5 RNA dependent RNA polymerase (RdRp) were proposed in the molecular docking? No have prior planning that supports the action of these compounds on these targets.

Response: 6-MMPR is a riboside analogue, so it was speculated its action would be by interfering with the RNA processing machinery of the virus. The two viral proteins that bind to RNA are NSP3-helicase domain and the NS5 RdRp domain.

Response to authors: Include this information in the manuscript.

3) In lines 86 to 88, describe that the compounds that will be evaluated were previously synthesized by the group and contain the cited heterocyclic.

Author Response

 Response to Reviewer 3 Comments

1) In lines 60 to 78, the authors cite several works described in the literature that do not justify the importance of choosing naphthoquinone, phtalimide and 1,2,3-triazole nuclei as pharmacophores in planning new hybrids.

Response: We presented a general description of heterocyclic compounds with anti-ZIKV activity. There are a limited anti-ZIKV activity reports in the literature from triazole.

Response to authors: The paragraph in lines 60-78 should be redone. It has no connection with the posterior paragraph. Please keep the explanation about the use of nucleosides and remove all other examples that are unrelated to the work.

Answer:  All suggested changes have been made.

 Cite the limited anti-ZIKV activity reports in the literature about 1,2,3-triazole and move the lines 82 to 86 to this paragraph. These lines explain about the choice of 1,4-naphthoquinone and phthalimide in the research.

Answer: All suggested changes have been made.

2) Why non-structural proteins NSP3 helicase and NSP5 RNA dependent RNA polymerase (RdRp) were proposed in the molecular docking? No have prior planning that supports the action of these compounds on these targets.

Response: 6-MMPR is a riboside analogue, so it was speculated its action would be by interfering with the RNA processing machinery of the virus. The two viral proteins that bind to RNA are NSP3-helicase domain and the NS5 RdRp domain.

Response to authors: Include this information in the manuscript.

Answer:  We have included this information in the manuscript (lines 234-238).

3) In lines 86 to 88, describe that the compounds that will be evaluated were previously synthesized by the group and contain the cited heterocyclic.

Answer:  All suggested changes have been made.
